# Canonical Autoregressive Generation

**Ivi Chatzi** [1] **Nina Corvelo Benz** [1,2] **Stratis Tsirtsis** [1] **Manuel Gomez-Rodriguez** [1]

## Abstract

State of the art large language models are trained using large amounts of tokens derived from raw text using what is called a tokenizer. Crucially, the tokenizer determines the (token) vocabulary a model will use during inference as well as, in principle, the (token) language. This is because, while the token vocabulary may allow for different tokenizations of a string, the tokenizer always maps the string to only one of these tokenizations—the canonical tokenization. However, multiple lines of empirical evidence suggest that large language models do not always generate canonical token sequences, and this comes with several negative consequences. In this work, we first show that, to generate a canonical token sequence, a model needs to generate (partial) canonical token sequences at each step of the autoregressive generation process underpinning its functioning. Building upon this theoretical result, we introduce canonical sampling, a simple and efficient sampling method that precludes a given model from generating non-canonical token sequences. Further, we also show that, in comparison with standard sampling, the distribution of token sequences generated using canonical sampling is provably closer to the true distribution of token sequences used during training.

## 1. Introduction

One of the most distinctive characteristics of large language models (LLMs) is that they are trained using large amounts of text data from diverse sources, including websites, books, scientific articles and code repositories (Naveed et al., 2024). During the training process, a tokenizer breaks down the text into sequences of tokens, which are the units that make up sentences and paragraphs, and the LLM learns to predict the next token in a sequence based on the preceding tokens.

Importantly, since tokenizers typically utilize a deterministic, rule-based mapping from text to token sequences (Gage, 1994; Sennrich et al., 2016; Kudo, 2018; Song et al., 2021), each unique string is always mapped into one tokenization referred to as the canonical tokenization (Geh et al., 2024). Consequently, one may conclude that, at the end of the training process, LLMs will have learned to generate exclusively canonical token sequences. However, there is empirical evidence that this conclusion does not hold—LLMs do generate both canonical and non-canonical token sequences (Cao & Rimell, 2021; Chirkova et al., 2023; Geh et al., 2024; Vieira et al., 2024; Giulianelli et al., 2024; Artola Velasco et al., 2025).

In this work, we argue that the generation of non-canonical token sequences is unlikely to bring performance benefits, and it does actually have negative consequences. First, it does not increase the expressiveness of a LLM since any non-canonical token sequence can be mapped to a canonical token sequence while preserving its string-level representation. However, they can be exploited to circumvent safety guidelines because they are out-of-distribution samples (Geh et al., 2025). Second, it can lead to a multiplicity of (plausible) tokenizations of an output string, and this creates a financial incentive for an LLM provider to strategize and misreport the (number of) tokens a model used to generate the output (Artola Velasco et al., 2025). Third, in the context of LLM evaluation, it becomes (computationally) hard to compute the perplexity of a given string (Cao & Rimell, 2021; Chirkova et al., 2023; Geh et al., 2024).

Motivated by the above observations, we develop a simple and efficient sampling method that precludes an LLM from generating non-canonical token sequences. Our method is based on the following theoretical result, which may be of independent interest: to generate a canonical token sequence, a model needs to generate (partial) canonical token sequences at each step of the autoregressive generation process underpinning its functioning. More specifically, building upon this result, our method utilizes the Gumbel-Max trick to efficiently sample only from the subset of tokens that can lead to (partial) canonical token sequences at each step of the generation. Importantly, we are able to show that,

---

[1]Max Planck Institute for Software Systems, Kaiserslautern, Germany [2]Department of Biosystems Science and Engineering, ETH Zurich, Basel, Switzerland. Correspondence to: Ivi Chatzi <ichatzi@mpi-sws.org>.

*Non-archival presentation at ICML 2025 Tokenization Workshop (TokShop)*, Vancouver, Canada. 2025.

in comparison with standard sampling, the distribution of token sequences generated by canonical sampling is provably closer to the true distribution of token sequences used during training.

**Related work.** The study of tokenization has a rich history in natural language processing (Palmer, 2000; Jurafsky & Martin, 2025). More recently, in the context of LLMs, there has been a renewed interest in formalizing tokenization and analyzing its properties (Gastaldi et al., 2024; Phan et al., 2024; Rajaraman et al., 2025), with the Byte-Pair Encoding (BPE) tokenization algorithm in particular receiving increased attention (Berglund & van der Merwe, 2023; Zouhar et al., 2024; Kozma & Voderholzer, 2024). The impact of tokenization on LLMs has also been studied empirically (Hou et al., 2023; Athiwaratkun et al., 2024) in generation tasks involving foreign languages (Fujii et al., 2023), translation (Domingo et al., 2019), arithmetic (Singh & Strouse, 2024), mental health (Liu et al., 2023), and privacy (Kharitonov et al., 2021; Petrov et al., 2023), among others. Moreover, there have also been efforts to design token-free LLMs that operate at the character or byte level (Clark et al., 2022; Tay et al., 2022; Xue et al., 2022; Yu et al., 2023; Wang et al., 2024), as well as stochastic tokenizers (Kudo, 2018; Provilkov et al., 2020). However, none of the above works has paid attention to the existence and significance of non-canonical tokenizations.

Only very recently, there has been a paucity of work studying the impact of non-canonical tokenizations on text perplexity calculations (Cao & Rimell, 2021; Chirkova et al., 2023; Geh et al., 2024; Vieira et al., 2024; Giulianelli et al., 2024), safety guidelines (Geh et al., 2025), and financial incentives of LLM providers (Artola Velasco et al., 2025). However, none of the above works has studied how to prevent the generation of non-canonical token sequences, which is the focus of our work. An exception is a very recent, independent work by Vieira et al. (2025) that also presents methods to ensure LLMs generate only canonical token sequences. However, their proposed sampling algorithm requires computing the full subset of tokens that lead to (partial) canonical token sequences at each step of the generation process, which is computationally inefficient compared to our approach that utilizes the Gumbel-Max trick. Additionally, Vieira et al. (2025) only investigate tokenizers based on the Byte-Pair Encoding tokenization algorithm, while we also consider Unigram- and Wordpiece-based tokenizers.

## 2. Preliminaries

In this section, we first define and formally characterize (deterministic) tokenizers and canonical tokenizations. Then, we briefly review the popular Byte-Pair Encoding (BPE) tokenization algorithm (Gage, 1994; Sennrich et al., 2016). Finally, we conclude with a description of the aspects of

LLM training and generation that are relevant for our work.

**Tokenizers and canonical tokenizations.** Tokenizers are tools that operate on sequences of characters (*i.e.*, strings) and sequences of tokens, and can transform one type into the other. Formally, let $\Sigma$ be a finite set of characters and $\Sigma^*$ be the set of all finite strings that can be created using the characters in $\Sigma$. Similarly, let $V$ be a finite set of tokens, which we will refer to as the vocabulary, and $V^*$ be the set of all finite token sequences that can be created using the tokens in $V$. Then, a tokenizer $\mathcal{T}$ is characterized by a tuple $\mathcal{T} \coloneqq (\Sigma, V, \mathtt{enc}, \mathtt{dec})$, where $\mathtt{enc} : \Sigma^* \to V^*$ is an encoder, which transforms strings to token sequences, and $\mathtt{dec} : V^* \to \Sigma^*$ is a decoder, which transforms token sequences to strings.

Let $\boldsymbol{\sigma}$ be a string and $\mathbf{s} \in V^*$ be a token sequence such that $\mathtt{dec}(\mathbf{s}) = \boldsymbol{\sigma}$, then, we will say that $\mathbf{s}$ is a (valid) tokenization of the string $\boldsymbol{\sigma}$. Here, note that there may be multiple tokenizations of a string $\boldsymbol{\sigma}$, that is, there may exist $\mathbf{s}, \mathbf{s}' \in V^*$ such that $\mathbf{s} \neq \mathbf{s}'$ and $\mathtt{dec}(\mathbf{s}) = \mathtt{dec}(\mathbf{s}') = \boldsymbol{\sigma}$. However, given a string $\boldsymbol{\sigma}$, the encoder deterministically picks a single tokenization $\mathtt{enc}(\boldsymbol{\sigma})$ among all tokenizations of $\boldsymbol{\sigma}$, which is often called the canonical tokenization (Geh et al., 2024).

**The BPE tokenization algorithm.** There exist many tokenization algorithms to construct the set of characters $\Sigma$, the set of tokens $V$, the encoder $\mathtt{enc}$, and the decoder $\mathtt{dec}$ characterizing a tokenizer $\mathcal{T}$. Here, we focus on the BPE tokenization algorithm (Gage, 1994; Sennrich et al., 2016), since it is the algorithm used by most, if not all, state-of-the-art LLMs.[1] In a nutshell, the BPE algorithm aims to create a tokenizer $\mathcal{T}$ with a set of tokens $V$ corresponding to character sequences that appear frequently in a training set of strings $\mathcal{C}$. To this end, it proceeds as follows.

In an initialization phase, the algorithm sets $\Sigma$ to be the set of all characters that appear at least once in $\mathcal{C}$, $V$ to be the set of single-character tokens, that is, for each $c \in \Sigma$, there exists one and only one $t \in V$ such that $\mathtt{dec}(t) = c$, and $\mathcal{S}$ to be the set of single-character token sequences $\mathbf{s} \in V^*$ representing all strings in $\mathcal{C}$. After the initialization phase, the algorithm proceeds iteratively for a predetermined number of iterations. At each iteration, it looks for the pair of tokens $t, t' \in V$ whose concatenation $t \mid t'$ appears most frequently in the set of token sequences $\mathcal{S}$, it creates a new token $t \circ t'$, where the symbol $\circ$ denotes the merge operation and $\mathtt{dec}(t \circ t') = \mathtt{dec}(t) \mid \mathtt{dec}(t')$, and it adds the newly created token to $V$. Then, for each token sequence $\mathbf{s} \in \mathcal{S}$, it replaces all occurrences of $t \mid t'$ by $t \circ t'$ one by one. Lastly, it creates a merge rule $r_{t,t'}$, which specifies the replacement

---

[1]In Appendix B, we review Unigram (Kudo, 2018) and Wordpiece (Song et al., 2021), which are also well-known tokenization algorithms.

of $t \mathbin{|} t'$ with $t \circ t'$, and adds it to an ordered sequence of merge rules $\mathcal{R}$.

After termination, the algorithm defines the encoder `enc` and decoder `dec` as follows. For any given token sequence $\mathbf{s} \in V^*$, $\mathtt{dec}(\mathbf{s})$ transforms the sequence to a string one token at a time, in order, using the token definitions. For any given string $\boldsymbol{\sigma} \in \Sigma^*$, $\mathtt{enc}(\boldsymbol{\sigma})$ first transforms the string to a sequence of single-character tokens. Then, it merges consecutive tokens from this sequence following the merge rules from $\mathcal{R}$, in order, until no merge rule is applicable, and it returns the resulting sequence—the canonical sequence.[2]

**LLM training and generation.** During training, an LLM learns to predict the next-token in canonical sequences of tokens derived from raw text using a tokenizer. More formally, given a (partial) token sequence $\mathbf{s} \in V^*$, the goal is (typically) to minimize the (cross-entropy) loss between the model's predicted distribution $d_\mathbf{s} \in \Delta(V)$ and the true next-token distribution $p_\mathbf{s} = P[T \mid \boldsymbol{S} = \mathbf{s}]$.

During generation, an LLM takes as input a prompt sequence $s_q \in V^*$ and responds with an output sequence $s \in V^*$, generated using an autoregressive process. At each time step of the process, the LLM first takes as input the concatenation of the prompt sequence $s_q$ and the (partial) output sequence $s$, and generates a distribution over tokens $d_{s_q|s} \in \Delta(V)$. Then, it samples the next token $t$ from the distribution $d_{s_q|s}$ and appends the token $t$ to the output sequence $s$. The process continues until a special end-of-sequence token is sampled.

Importantly, since LLMs are trained on finite data, the support of the distribution $d_{\mathbf{s}_q|\mathbf{s}}$ may differ from the respective true distribution $p_{\mathbf{s}_q|\mathbf{s}}$. As a result, it is possible for an LLM to generate a non-canonical token sequence, even if it has encountered no such sequences during training (Cao & Rimell, 2021; Chirkova et al., 2023; Geh et al., 2024; Vieira et al., 2024; Giulianelli et al., 2024; Geh et al., 2025; Artola Velasco et al., 2025).

# 3. An Efficient Approach to Canonical Autoregressive Generation

In this section, we start by showing that, for an output token sequence generated by an LLM to be canonical, the partial token sequences generated at each step of the autoregressive generation must also be canonical. Building upon this result, we then introduce canonical sampling, a next-token sampling method that enables an LLM to generate only canonical token sequences, as well as an efficient implementation of canonical sampling using the Gumbel-Max trick (Huijben et al., 2022). Finally, we conclude by

showing that, in comparison with standard sampling, the distribution of token sequences generated using canonical sampling is provably closer to the true distribution of token sequences used during training.

## 3.1. Subsequences of Canonical Token Sequences Must Also Be Canonical

Our starting point is the following theorem, which tells us that, if an LLM generates a non-canonical (partial) output sequence at any step of the autoregressive generation process, the output sequence is bound to remain non-canonical in subsequent steps:[3]

**Theorem 3.1.** *Let* $\mathcal{T} = (\Sigma, V, \mathtt{enc}, \mathtt{dec})$ *be a BPE-based tokenizer[4] and* $\mathbf{s} \in V^*$ *a non-canonical token sequence according to* $\mathcal{T}$. *Then, for any token* $t \in V$, $\mathbf{s} \mathbin{|} t$ *is also non-canonical.*

*Proof sketch.* We prove the theorem by contradiction. We first assume that there exists a non-canonical sequence $\mathbf{s}$ and a token $t$ such that $\mathbf{s} \mathbin{|} t$ is canonical. This implies that, applying in order the merge rules of the tokenizer's merge rule sequence $\mathcal{R}$, to the single-character token sequence corresponding to the string $\mathtt{dec}(\mathbf{s} \mathbin{|} t)$, results in the token sequence $\mathbf{s} \mathbin{|} t$ itself. We then show that the applied merges can be split into two separate groups that would independently result in $\mathbf{s}$ and $t$ starting from the corresponding strings $\mathtt{dec}(\mathbf{s})$ and $\mathtt{dec}(t)$, respectively. Based on that, we argue that, since $\mathbf{s}$ is non-canonical, there must exist a different, canonical tokenization $\mathbf{s}' \neq \mathbf{s}$ of the string $\mathtt{dec}(\mathbf{s})$, created by applying a different sequence of merge rules than the one that resulted in $\mathbf{s}$. The rest of the proof is based on the observation that, since the merges that create the last token $t$ while transforming $\mathtt{dec}(\mathbf{s} \mathbin{|} t)$ into $\mathbf{s} \mathbin{|} t$ are independent of the merges that transform $\mathtt{dec}(\mathbf{s})$ into $\mathbf{s}$, the string $\mathtt{dec}(\mathbf{s})$ should have been tokenized following the merge rules that result in $\mathbf{s}'$, which contradicts the fact that $\mathbf{s} \mathbin{|} t$ is canonical. $\square$

The above theorem readily implies that an output token sequence generated by an LLM is canonical if and only if the partial token sequences generated at each step of the autoregressive generation process are canonical. This theorem also provides a plausible explanation to the empirical observation that the likelihood that an LLM generates non-canonical output sequences increases with the length of the sequence (Geh et al., 2024). This is because, since sampling a "non-canonical token" once during the autoregressive process is sufficient to render the output token sequence non-canonical, it is natural that the chances of this to happen increase with the number of sampled tokens.

---

[2]If $t \mathbin{|} t'$ appears multiple times in a token sequence, the merge rule $r_{t,t'}$ is applied in order of appearance in the sequence.

[3]All proofs can be found in Appendix A.

[4]Our results generalize to Unigram- and Wordpiece-based tokenizers, which we discuss further in Appendix B.

## 3.2. Canonical Sampling

At each step of the autoregressive generation process, canonical sampling sets the probability of a subset of tokens to zero—those that, when appended to a partial output sequence, would result in a non-canonical token sequence—and redistributes their probability mass to the remaining tokens proportionally to their original probability mass.

Formally, let $d_\mathbf{s}$ denote the next-token distribution generated by the LLM given a partial output token sequence $\mathbf{s}$ and let $d_\mathbf{s}(t)$ denote the probability of sampling a token $t$ from this distribution. Given the partial output token sequence $\mathbf{s}$, an LLM using canonical sampling draws the next token in the autoregressive generation process from a *canonicalized* next-token distribution

$$\tilde{d}_\mathbf{s}(t) := \begin{cases} d_\mathbf{s}(t)/Z & \text{if } \mathbf{s} \mid t \text{ is canonical} \\ 0 & \text{otherwise,} \end{cases} \qquad (1)$$

where $Z = \sum_{t \in V:\ \mathbf{s} \mid t \text{ is canonical}} d_\mathbf{s}(t)$ is a normalization constant that ensures that $\tilde{d}_\mathbf{s}$ is a valid probability distribution. In that context, note that redistributing the probability mass of tokens that would lead to non-canonical token sequences proportionally to the original probabilities $d_\mathbf{s}(t)$ is a natural choice we make, inspired by other popular sampling methods used by LLMs, such as top-$k$ and top-$p$ sampling (Holtzman et al., 2019).

To compute the canonicalized next-token distribution $\tilde{d}_\mathbf{s}$ and sample from it directly, one would need to evaluate whether the token sequence $\mathbf{s} \mid t$ is canonical for every token $t \in V$, which can be computationally expensive, especially since these evaluations are required at every step of the autoregressive generation process. Fortunately, we can efficiently sample from the canonicalized next-token distribution $\tilde{d}_\mathbf{s}$ using Algorithm 1.

Algorithm 1 starts by sampling a value $u_t \sim \text{Gumbel}(0,1)$ from a Gumbel distribution for each token $t \in V$. Then, it ranks the tokens in decreasing order with respect to the perturbed log-probability $\log(d_\mathbf{s}(t)) + u_t$. Finally, it returns the token $t$ with the largest value of $\log(d_\mathbf{s}(t)) + u_t$ such that $\mathbf{s} \mid t$ is canonical. Algorithm 1 relies on a property of the Gumbel-Max trick (Maddison et al., 2014; Huijben et al., 2022), which states that the argmax operation over a constrained subset of categorical outcomes is equivalent to sampling from a categorical distribution with zero probability for all outcomes outside the subset, and with the probabilities of the outcomes in the subset scaled proportionally to their original probabilities, as shown in Eq. 2 in Maddison et al. (2014). Hence, it readily holds that Algorithm 1 returns a valid sample from the canonicalized next-token distribution $\tilde{d}_\mathbf{s}$ defined in Eq. 1, *i.e.*,

$$\underset{t \in V:\ \mathbf{s} \mid t \text{ is canonical}}{\text{argmax}} \{\log(d_\mathbf{s}(t)) + u_t\} \sim \tilde{d}_\mathbf{s}.$$

---

**Algorithm 1** Canonical Sampling with Gumbel-Max

**Require:** next-token distribution $d_\mathbf{s}$
    $u_t \sim Gumbel(0,1)$ for all $t \in V$
    **for** $t \in V$ in decreasing order of $\log(d_\mathbf{s}(t)) + u_t$ **do**
        **if** $s \mid t$ is canonical **then**
            **return** t
        **end if**
    **end for**

---

Further, it is worth highlighting that, in contrast to computing the canonicalized next-token distribution $\tilde{d}_\mathbf{s}$, which requires evaluating the canonicity of $|V|$ token sequences, Algorithm 1 requires only a few evaluations of canonicity. This is because, in practice, LLMs tend to generate mostly canonical token sequences (Geh et al., 2024), hence, the probabilities $d_\mathbf{s}(t)$ generated by an LLM for tokens $t$ that lead to non-canonical sequences $\mathbf{s} \mid t$ are usually small. More specifically, let $d_\mathbf{s}(\text{canonical})$ be the probability mass on the subset of tokens that lead to canonical sequences, *i.e.*, $d_\mathbf{s}(\text{canonical}) = \sum_{t \in V:\ \mathbf{s} \mid t \text{ is canonical}} d_\mathbf{s}(t)$, then Algorithm 1 requires, in expectation, fewer than $1/d_\mathbf{s}(\text{canonical})$ evaluations of canonicity before successfully sampling the next token. That is because, unlike (independent) rejection sampling from $d_\mathbf{s}$, which would require in expectation exactly $1/d_\mathbf{s}(\text{canonical})$ evaluations of canonicity until a token that leads to a canonical sequence is successfully sampled, our approach never checks the same token twice, which results in an increase in the success probability of sampling a token that leads to a canonical sequence after each failed attempt.[5]

### 3.3. Canonical Sampling Brings us Closer to the True Distribution of Token Sequences

Let $p$ denote the true distribution over token sequences $\mathbf{s} \in V^*$ in the (token) language used during training, for which note that it holds that $p(\mathbf{s}) = 0$ for all sequences $\mathbf{s}$ that are non-canonical. Moreover, let $d$ denote the distribution over token sequences that the LLM generates using standard sampling, and $\tilde{d}$ the distribution over token sequences that the LLM generates using canonical sampling, that is, sampling from the canonicalized next token distribution $\tilde{d}_\mathbf{s}$ given by Eq. 1 at each step of the autoregressive generation process. Then, the following theorem shows that $p$ is likely to be closer to $\tilde{d}$ than $d$ in terms of KL-divergence:

---

[5]The number of evaluations of canonicity in rejection sampling is distributed according to a geometric distribution with success probability $d_\mathbf{s}(\text{canonical})$ resulting in $1/d_\mathbf{s}(\text{canonical})$ evaluations in expectation until a successful sample.

**Theorem 3.2.** *Let $d$ be absolutely continuous[6] with respect to $p$. Moreover, assume that there exist $\mathbf{s} \in V^*$ and $t_1, t_2 \in V$ such that $\mathbf{s} \mid t_1$ is non-canonical with $d(\mathbf{s} \mid t_1) > 0$ and $\mathbf{s} \mid t_2$ is canonical with $p(\mathbf{s} \mid t_2) > 0$ and $d(\mathbf{s} \mid t_2) > 0$. Then, it holds that*

$$KL(p, \tilde{d}) < KL(p, d). \tag{2}$$

In words, the two assumptions for the theorem to hold are that (i) there must exist non-canonical token sequences with positive probability of being generated under $d$ so that their probability mass can be redistributed, and (ii) there must exist canonical token sequences with positive probability under $d$ and $p$ so that the redistribution of probability mass in $\tilde{d}$ is beneficial. To understand the intuition behind Theorem 3.2, note that, by using canonical sampling (*i.e.*, sampling from $\tilde{d}$ instead of $d$), the probability that an LLM generates non-canonical token sequences becomes zero, and the probability that it generates any other (canonical) token sequence increases under $\tilde{d}$. Further, since only canonical token sequences have positive probability under the true distribution $p$, this redistribution of probability mass from non-canonical token sequences to canonical ones can only bring the distribution $\tilde{d}$ closer to the true distribution $p$ compared to $d$.

On the flip side, it is important to clarify that a similar property does not necessarily hold for the respective distributions over strings. That is, using canonical sampling, the distribution of output strings, resulting from decoding the output token sequences, is not guaranteed to be closer (in terms of KL-divergence) to the true distribution of output strings used during training. Formally, let $p_{\text{dec}} = P_{\mathbf{s} \sim p(\mathbf{s})}[\text{dec}(\mathbf{s})]$ be the true distribution over strings, $d_{\text{dec}} = P_{\mathbf{s} \sim d(\mathbf{s})}[\text{dec}(\mathbf{s})]$ be the distribution of strings induced by the distribution of output token sequences $d$, and $\tilde{d}_{\text{dec}} = P_{\mathbf{s} \sim \tilde{d}(\mathbf{s})}[\text{dec}(\mathbf{s})]$ be the distribution of strings induced by the distribution of output token sequences $\tilde{d}$. Then, we can show that

$$KL(p_{\text{dec}}, \tilde{d}_{\text{dec}}) = KL(p, \tilde{d}) \tag{3}$$

because there is a one-to-one mapping determined by the encoder `enc` from any string to a canonical token sequence, and only canonical token sequences have positive probability under $p$ and $\tilde{d}$. In contrast, one cannot claim the same for $KL(p_{\text{dec}}, d_{\text{dec}})$ and $KL(p, d)$, as the same string can have multiple tokenizations that have positive probability under $d$. Thus, Theorem 3.2 does not imply that $KL(p_{\text{dec}}, \tilde{d}_{\text{dec}}) < KL(p_{\text{dec}}, d_{\text{dec}})$.

---

[6]Absolute continuity is required for the KL-divergence to be well defined, *i.e.*, we require that $d(\mathbf{s}) = 0$ implies that $p(\mathbf{s}) = 0$ for all $\mathbf{s} \in V^*$.

## 4. Discussion and Future Work

In this section, we mention aspects of our work that we believe deserve additional consideration and discuss avenues for future work.

**Methodology.** Our main theoretical result reveals that, for BPE-, Unigram- and Wordpiece-based tokenizers, subsequences of canonical token sequences must also be canonical. This is somehow surprising since these (deterministic) tokenizers use fundamentally different tokenization techniques—BPE uses a rule based approach, Unigram maximizes the probability of the tokenization, and Wordpiece greedily encodes the text to minimize the number of tokens. Consequently, it would be very interesting to better understand what property a tokenizer needs to satisfy for our main theoretical result to hold. In this context, it would also be interesting to define relaxed notions of canonical tokenization applicable to stochastic tokenizers and adapt our main theoretical result to this type of tokenizers. Further, it would be important to extend the formalization of canonical tokenization, and, consequently, our theoretical results and sampling method, to account for the potential impact of pre-tokenization, a process that partitions the input text before the tokenization algorithm is applied separately to each partition.

Under canonical sampling, we canonicalize the next-token distribution by redistributing the probability mass of tokens leading to non-canonical token sequences among the remaining tokens proportionally to their original probability mass. We have shown that, in comparison with the original next-token distribution, this particular canonicalized next-token distribution leads to a distribution of output sequences that is closer to the true distribution of token sequences. However, there may be other canonicalized next-token distribution leading to a distribution of output sequences that are even closer to the true distribution. In future work, it would be worth to compare different canonicalized next-token distributions, and investigate global strategies beyond (next-token) sampling to optimally redistribute the probability mass of non-canonical output token sequences.

**Evaluation.** As discussed at the end of Section 3.3, we cannot establish whether Theorem 3.2, which focuses on distributions of output token sequences, can be extended meaningfully to distributions of output strings. This limitation arises because the distribution of output strings inherently marginalizes across all possible tokenizations of any given string, creating a complex relationship between token- and string-level probability distributions. Consequently, the KL-divergence between the true distribution of strings and the distribution of output strings induced by the original next-token distribution remains theoretically intractable when compared to the corresponding KL-divergence com-

puted in token space. Given this theoretical gap, it would be important to evaluate to which extent canonical sampling offers practical performance benefits across different state-of-the-art LLMs, particularly since users typically derive value from the string that the output token sequence represents, rather than the token sequence itself.

## 5. Conclusions

We have proposed canonical sampling, a simple and efficient sampling method based on the Gumbel-Max trick that ensures a given LLM generates canonical token sequences at each step of the autoregressive generation process underpinning its functioning. Further, we have shown that, in comparison with standard sampling, the distribution of token sequences generated using canonical sampling is provably closer to the true distribution of token sequences used during training.

## Acknowledgements

Gomez-Rodriguez acknowledges support from the European Research Council (ERC) under the European Union's Horizon 2020 research and innovation programme (grant agreement No. 945719).

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

# A. Proof of Theorem 3.1

Here, we define some additional notation regarding the BPE tokenization algorithm and prove Theorem 3.1.

When tokenizing a string $\sigma = c_1 \mid \cdots \mid c_n$, with $c_i \in \Sigma$ and $n \in \mathbb{N}$, according to the BPE algorithm, we use the term *merge* and write $m = (r_{t,t'}, i, j)$ to refer to a single application of merge rule $r_{t,t'} \in \mathcal{R}$ on two consecutive tokens $t \mid t'$ that correspond to the substring of characters $c_i \mid \cdots \mid c_j$ in $\sigma$. To tokenize $\sigma$, merges are performed following a unique merge sequence $M = (m_1, \ldots, m_{|M|})$, where the merges are ordered $m_1 \prec \cdots \prec m_{|M|}$, first by the order in which the merge rule they refer to appears in $\mathcal{R}$, and second by position of merged token pairs in the sequence. The notation $m \prec m'$, for $m = (r, i, j), m' = (r', i', j')$ with $r, r' \in \mathcal{R}$, $i, j, i', j' \in [n]$, means that either $r$ appears before $r'$ in $\mathcal{R}$, or $r = r'$ and $i < i'$.

We now define an operator that, applied to a merge sequence $M$ that tokenizes the string $\sigma$, specifies the subsequence of merges that are applied to a certain substring of $\sigma$. Further, we define *shift equivalence*, referring to merge sequences whose merges correspond to the exact same merge rule sequence applied to different positions in a string (shifted by a constant).

**Definition A.1.** Let $\mathbf{s} = t_1 \mid \cdots \mid t_{|\mathbf{s}|} \in V^*$ be a tokenization of $\sigma = c_1 \mid \cdots \mid c_{|\sigma|} \in \Sigma^*$ obtained by applying merge sequence $M = (m_1, \ldots, m_n)$. For any continuous token subsequence $\mathbf{s}'$ of $\mathbf{s}$ spanning the substring $\sigma' = c_u \mid \cdots \mid c_v$, $1 \leq u < v \leq |\sigma|$, the operator $[M]_{\mathbf{s}'}$ denotes the subsequence of merges in $M$ such that $m = (r, i, j) \in [M]_{\mathbf{s}'}$ if $m \in M$ and $u \leq i < j \leq v$.

**Definition A.2.** Two merge sequences $M = (m_1, \ldots, m_{|M|})$, $M' = (m'_1, \ldots, m'_{|M'|})$ are *shift equivalent*, denoted by $M \stackrel{\rightarrow}{\equiv} M'$, if $|M| = |M'|$ and there exists $n \in \mathbb{Z}$ such that for all $i \in \{1, \ldots, |M|\}$ with $m_i = (r, j, k), r \in \mathcal{R}, k > j > 0$, it holds that $m'_i = (r, j + n, k + n)$.

Before we prove Theorem 3.1, we show that the merge sequence that creates the tokenization $\mathbf{s} = \mathbf{s}_1 \mid \cdots \mid \mathbf{s}_n$ from a string $\sigma$, can be partitioned into $n$ disjoint (non-continuous) subsequences of merges, that create the tokenizations $\mathbf{s}_1, \ldots, \mathbf{s}_n$ from the corresponding substring of $\sigma$.

**Lemma A.3.** *Let $\mathbf{s} \in V^*$ be a tokenization of $\sigma = c_1 \mid \cdots \mid c_{|\sigma|} \in \Sigma^*$ obtained by applying merge sequence $M_{\mathbf{s}}$. For any partition $\mathbf{s} = \mathbf{s}_1 \mid \cdots \mid \mathbf{s}_n$, where $\mathbf{s}_i \in V^*, i \in [n], n \in \mathbb{N}$, the following hold:*

1. *For each $\mathbf{s}_i \in \mathbf{s}$, there exists a merge sequence $M_{\mathbf{s}_i}$ such that applying $M_{\mathbf{s}_i}$ to the string $\mathrm{dec}(\mathbf{s}_i)$ creates $\mathbf{s}_i$ and $[M_{\mathbf{s}}]_{\mathbf{s}_i} \stackrel{\rightarrow}{\equiv} M_{\mathbf{s}_i}$.*

2. *For all $\mathbf{s}_i, \mathbf{s}_j \in \mathbf{s}, i \neq j$, if $m \in [M_{\mathbf{s}}]_{\mathbf{s}_i}$ then $m \notin [M_{\mathbf{s}}]_{\mathbf{s}_j}$ and vice-versa,*

3. *For each merge $m \in M_{\mathbf{s}}$ there exists $\mathbf{s}_i \in \mathbf{s}$ such that $m \in [M_{\mathbf{s}}]_{\mathbf{s}_i}$.*

*Proof.* 1. If $\mathbf{s}_i$ is a tokenization of a single character $\mathrm{dec}(\mathbf{s}_i) = c$, then $[M_{\mathbf{s}}]_{\mathbf{s}_i}$ is the empty sequence and the statement holds trivially. Assume $\mathbf{s}_i$ is a tokenization of the substring $c_u \mid \cdots \mid c_v$ of $\sigma$, with $v > u > 0$, and $[M_{\mathbf{s}}]_{\mathbf{s}_i} = (m_1, \ldots, m_n)$. By Definition A.1, all merges $m = (r, j_1, j_2) \in M_{\mathbf{s}}$ with $u \leq j_1 < j_2 \leq v$ belong in $[M_{\mathbf{s}}]_{\mathbf{s}_i}$, so these merges tokenize $c_u \mid \cdots \mid c_v$ into $\mathbf{s}_i$. Then, the merge sequence $M_{\mathbf{s}_i} = (m'_1, \ldots, m'_n)$, where for all $k \in [n]$ it holds that $m_k = (r, j_1, j_2)$ and $m'_k = (r, j_1 - u, j_2 - u), r \in \mathcal{R}$, contains the same merge rules in the same order, but with indices shifted left by $u$. So if $M_{\mathbf{s}_i}$ is applied to the string $\mathrm{dec}(\mathbf{s}_i)$ it will create $\mathbf{s}_i$.

2. If $[M_{\mathbf{s}}]_{\mathbf{s}_i}$ or $[M_{\mathbf{s}}]_{\mathbf{s}_j}$ are the empty sequence, meaning $\mathbf{s}_i$ or $\mathbf{s}_j$ are a tokenization of only a single character in $\sigma$, then the statement holds trivially. If $\mathbf{s}_i$ is a tokenization of the substring $c_u \mid \cdots \mid c_v$ and $\mathbf{s}_j$ is a tokenization of the substring $c_{u'} \mid \cdots \mid c_{v'}$, since $i \neq j$ is must be that either $u < v < u' < v'$ or $u' < v' < u < v$. But for all $m = (r, i_1, i_2) \in [M_{\mathbf{s}}]_{\mathbf{s}_i}$ it holds that $u \leq i_1 < i_2 \leq v$, and for all $m' = (r', j_1, j_2) \in [M_{\mathbf{s}}]_{\mathbf{s}_j}$ it holds that $u' \leq j_1 < j_2 \leq v'$. Intuitively, it is not possible for a merge to span two different subsequences $\mathbf{s}_i, \mathbf{s}_j$ in the partition of $\mathbf{s}$, because then (part of) $\mathbf{s}_i$ and $\mathbf{s}_j$ would be merged.

3. Each merge $m = (r, j, k) \in M_{\mathbf{s}}, r \in \mathcal{R}$ must have $1 \leq j < k \leq |\sigma|$. Because the whole string $\sigma$ is tokenized into $\mathbf{s}$ and, by definition, the token merged by $m$ cannot be part of two different subsequences in the partition, there must exist $\mathbf{s}_i \in \mathbf{s}$ that is a tokenization of a substring $c_u \mid \cdots \mid c_v$ of $\sigma$ with $u \leq j < k \leq v$. So by Definition A.1, $m \in [M_{\mathbf{s}}]_{\mathbf{s}_i}$.

$\square$

We can now prove Theorem 3.1, which we restate below.

**Theorem 3.1.** Let $\mathcal{T} = \{\Sigma, V, \texttt{enc}, \texttt{dec}\}$ be a BPE-based tokenizer and $\mathbf{s} \in V^*$ a non-canonical token sequence according to $\mathcal{T}$. Then, for any token $t \in V$, $\mathbf{s} \mid t$ is also non-canonical.

*Proof.* Assume that $\mathbf{s} \mid t$ is canonical. Then, there exists a unique merge sequence $M_{\mathbf{s} \mid t}$ that creates it following the BPE algorithm. From Lemma A.3, $M_{\mathbf{s} \mid t}$ can be split into $[M_{\mathbf{s} \mid t}]_{\mathbf{s}}$ and $[M_{\mathbf{s} \mid t}]_t$, where $[M_{\mathbf{s} \mid t}]_{\mathbf{s}}$ contains the merges that create $\mathbf{s}$ and $[M_{\mathbf{s} \mid t}]_t$ contains the merges that create $t$. From Lemma A.3, there exists a merge sequence $M_{\mathbf{s}}$ that creates $\mathbf{s}$ when applied to $\texttt{dec}(\mathbf{s})$ and $[M_{\mathbf{s} \mid t}]_{\mathbf{s}} \overset{\rightarrow}{\equiv} M_{\mathbf{s}}$. Because $\mathbf{s}$ is a prefix of $\mathbf{s} \mid t$, the index shift is zero and we have that $[M_{\mathbf{s} \mid t}]_{\mathbf{s}} = M_{\mathbf{s}}$.

Since $\mathbf{s}$ tokenized according to $[M_{\mathbf{s} \mid t}]_{\mathbf{s}} = (m_1, \ldots, m_n)$ is non-canonical, there must exist a different, canonical tokenization $\mathbf{s}' \neq \mathbf{s}$ of the same character string, $\texttt{dec}(\mathbf{s}) = \texttt{dec}(\mathbf{s}')$. Let $M_{\mathbf{s}'} = (m'_1, \ldots, m'_{n'})$ be the unique merge sequence that creates $\mathbf{s}'$ from $\texttt{dec}(\mathbf{s})$ according to the BPE algorithm. Because $M_{\mathbf{s}'} \neq [M_{\mathbf{s} \mid t}]_{\mathbf{s}}$, it must be that either there exists at least one $i$, $i \leq \min(n, n')$, such that $m_i \neq m'_i$, or $m_i = m'_i$ for all $i \in [\min(n, n')]$ but $n \neq n'$.

We will first examine the first case. Let $m_i \in [M_{\mathbf{s} \mid t}]_{\mathbf{s}}$ and $m'_i \in M_{\mathbf{s}'}$ be the first merges that are different between $[M_{\mathbf{s} \mid t}]_{\mathbf{s}}$ and $M_{\mathbf{s}'}$, meaning $\forall j < i: m_j = m'_j$, for $m_j \in [M_{\mathbf{s} \mid t}]_{\mathbf{s}}, m'_j \in M_{\mathbf{s}'}$. Because $\mathbf{s}'$ is canonical and $\mathbf{s}$ is not, it must be that $m'_i \prec m_i$. We will now compare $M_{\mathbf{s} \mid t}, [M_{\mathbf{s} \mid t}]_{\mathbf{s}}$ and $M_{\mathbf{s}'}$. There are two sub-cases:

1. The first $i$ merges in $M_{\mathbf{s} \mid t}$ are the same as in $[M_{\mathbf{s} \mid t}]_{\mathbf{s}}$. This means that the first $i - 1$ merges are the same as in $M_{\mathbf{s}'}$. Then, merge $m_i$ being applied instead of $m'_i \prec m_i$ on substring $\texttt{dec}(\mathbf{s})$, implies that $M_{\mathbf{s} \mid t}$ cannot be the merge sequence that creates the canonical tokenization of $\texttt{dec}(\mathbf{s} \mid t)$ according to BPE.

2. The first $i$ merges in $M_{\mathbf{s} \mid t}$ are not the same as in $[M_{\mathbf{s} \mid t}]_{\mathbf{s}}$. This means that there exists at least one merge $m \in M_{\mathbf{s} \mid t}$ among the first $i$ merges in $M_{\mathbf{s} \mid t}$ such that $m \notin [M_{\mathbf{s} \mid t}]_{\mathbf{s}}$. For any such merge $m$ as $m \notin [M_{\mathbf{s} \mid t}]_{\mathbf{s}}$, it must hold that $m \in [M_{\mathbf{s} \mid t}]_t$ by Lemma A.3. So, in $M_{\mathbf{s} \mid t}$, merge $m_i$ is preceded by the first $i - 1$ merges of $[M_{\mathbf{s} \mid t}]_{\mathbf{s}}$ and merge $m$. By Lemma A.3, $m$ does not affect the tokens that will create $\mathbf{s}$, so the only merges in $M_{\mathbf{s} \mid t}$ before $m_i$ that affect $\mathbf{s}$ are the first $i - 1$ merges of $[M_{\mathbf{s} \mid t}]_{\mathbf{s}}$, which are the same as $M_{\mathbf{s}'}$. Then, as in case 1, merge $m_i$ being applied instead of $m'_i \prec m_i$, implies that $M_{\mathbf{s} \mid t}$ cannot be the merge sequence that creates the canonical tokenization of $\texttt{dec}(\mathbf{s} \mid t)$ according to BPE.

We will now examine the case where $m_i = m'_i$ for all $i \in [\min(n, n')]$, $m_i \in [M_{\mathbf{s} \mid t}]_{\mathbf{s}}, m'_i \in M_{\mathbf{s}'}$ but $n \neq n'$. If $n > n'$, then there exists at least one merge that can be applied on $\mathbf{s}'$ after all merges of $M_{\mathbf{s}'}$ are done, which means that $\mathbf{s}'$ cannot be canonical. If $n' > n$, then there exists at least one merge that can be applied on $\mathbf{s}$ after all merges of $[M_{\mathbf{s} \mid t}]_{\mathbf{s}}$ are done. This merge can also be applied on $\mathbf{s} \mid t$, which means that $\mathbf{s} \mid t$ cannot be canonical.

All cases lead to a contradiction, which implies that $\mathbf{s} \mid t$ is non-canonical. $\qquad \square$

## B. Unigram- and Wordpiece-based Tokenizers

Here, we review the Unigram (Kudo, 2018) and Wordpiece (Song et al., 2021) tokenization algorithms, and prove that appending any token to a non-canonical token sequence results in a sequence that is also non-canonical, according to tokenizers based on these algorithms.

### B.1. The Unigram tokenization algorithm

The Unigram algorithm aims to create a tokenizer $\mathcal{T}$ with a set of tokens $V$ in order to minimize a loss when tokenizing a training set of strings $\mathcal{C}$. In the initialization phase, $\Sigma$ is set to contain all the characters that appear at least once in $\mathcal{C}$. Unlike the BPE algorithm, which iteratively adds tokens to the vocabulary $V$, Unigram starts with a large vocabulary and removes tokens from it until it reaches a predetermined size, specified by the LLM developer. This initial large vocabulary can be set in multiple ways, such as applying the BPE algorithm on $\mathcal{C}$ with many iterations, or initializing it with tokens that decode to the most frequently occuring substrings in $\mathcal{C}$.

After the initial vocabulary has been set, the algorithm proceeds in iterations, each time computing a loss over the strings in $\mathcal{C}$ and the current vocabulary, and removing a batch of tokens from the vocabulary (typically $10\%$ or $20\%$ of tokens) whose removal minimizes this loss. In each iteration, every token $t$ in the current vocabulary $V$ is assigned a probability score $r(t) = \frac{freq(t)}{\sum_{t' \in V} freq(t')}$, where $freq(t)$ denotes the number of times that the token $t$ appears in all possible tokenizations

of the strings in $\mathcal{C}$. For each token $t \in V$, the loss over the training set is computed as $\sum_{\boldsymbol{\sigma} \in \mathcal{C}} -\log(r_{V \setminus \{t\}}(\boldsymbol{\sigma}))$, where $r_V(\boldsymbol{\sigma}) = \max_{\mathbf{s} \in V^*, \text{dec}(\mathbf{s}) = \boldsymbol{\sigma}} r(\mathbf{s})$ denotes the probability score of the most likely tokenization of $\boldsymbol{\sigma}$ under vocabulary $V$, and the probability score of tokenization $\mathbf{s} = t_1 \mathbin{\text{\tiny|}} \cdots \mathbin{\text{\tiny|}} t_n$, with $n \in \mathbb{N}$, is simply $r(\mathbf{s}) = r(t_1) \ldots r(t_n)$. The tokens that minimize this loss are removed from the vocabulary and the process repeats until the vocabulary reaches a predetermined size.

After the vocabulary has been finalized, the encoder is set to tokenize a string $\boldsymbol{\sigma} \in \Sigma^*$ by finding its most likely tokenization under the final vocabulary $V$, *i.e.*, $\text{enc}(\boldsymbol{\sigma}) = \arg\max_{\mathbf{s} \in V^*, \text{dec}(\mathbf{s}) = \boldsymbol{\sigma}} r(\mathbf{s})$, using the Viterbi algorithm (Viterbi, 1967), and the decoder decodes all tokens in $V$ the same way as in the original, large vocabulary.

**Theorem B.1.** *Let $\mathcal{T} = (\Sigma, V, \text{enc}, \text{dec})$ be a Unigram-based tokenizer and $\mathbf{s} \in V^*$ a non-canonical token sequence according to $\mathcal{T}$. Then, for any token $t \in V$, $\mathbf{s} \mathbin{\text{\tiny|}} t$ is also non-canonical.*

*Proof.* If $\mathbf{s}$ is non-canonical according to Unigram, then let $\mathbf{s}'$ denote the canonical tokenization of the same character string, $\text{dec}(\mathbf{s}) = \text{dec}(\mathbf{s}')$. Because $\mathbf{s}'$ is canonical, it must be that $r(\mathbf{s}') > r(\mathbf{s})$. It follows that $r(\mathbf{s} \mathbin{\text{\tiny|}} t) = r(\mathbf{s})r(t) < r(\mathbf{s}')r(t) = r(\mathbf{s}' \mathbin{\text{\tiny|}} t)$, so $\mathbf{s} \mathbin{\text{\tiny|}} t$ cannot be the canonical tokenization of $\text{dec}(\mathbf{s} \mathbin{\text{\tiny|}} t)$. $\qquad\square$

## B.2. The Wordpiece tokenization algorithm

The Wordpiece algorithm is similar to BPE, in the sense that it builds the token vocabulary by iteratively merging tokens. However, the initialization phase, the merging criterion and the encoding function differ.

In the initialization phase, $\Sigma$ is set to contain all characters that appear at least once in the training set of strings $\mathcal{C}$. Then, for each character $c \in \Sigma$ that appears at least once in $\mathcal{C}$, a single-character token $t$ is added to $V$ such that $\text{dec}(t) = c$, and $\mathcal{S}$ is initialized as a set that contains all single-character token sequences $\mathbf{s} \in V^*$ that represent all strings in $\mathcal{C}$. Interestingly, Wordpiece transforms characters (and substrings) inside words differently than characters (and substrings) at the beginning of words. Specifically, tokens representing characters (and substrings) inside words have a special prefix.

To build the vocabulary, Wordpiece proceeds iteratively by merging existing tokens and adding them to $V$ until it reaches a predetermined size, similarly to BPE. However, the criterion to select which pair of tokens to merge is different. If $freq(\mathbf{s}')$ denotes the number of times that sequence $\mathbf{s}' \in V^*$ appears (as a subsequence) in the set of sequences $\mathcal{S}$, Wordpiece looks for the pair of tokens $t, t' \in V$ that maximizes the value of $\frac{freq(t \mathbin{\text{\tiny|}} t')}{freq(t) \cdot freq(t')}$. Then, a new token $t \circ t'$ is added to $V$, where $\text{dec}(t \circ t') = \text{dec}(t) \mathbin{\text{\tiny|}} \text{dec}(t')$, and all occurences of $t \mathbin{\text{\tiny|}} t'$ in each token sequence $\mathbf{s} \in \mathcal{S}$ are replaced by $t \circ t'$. With this criterion, Wordpiece prefers to merge tokens whose concatenation appears commonly in $\mathcal{S}$, but they are not common individually.

After the above iterative process terminates, the algorithm defines the encoder and decoder functions as follows. For any token sequence $\mathbf{s} = t_1 \mathbin{\text{\tiny|}} \cdots \mathbin{\text{\tiny|}} t_n \in V^*$ with $n \in \mathbb{N}$, the decoder returns $\text{dec}(\mathbf{s}) = \text{dec}(t_1) \mathbin{\text{\tiny|}} \cdots \mathbin{\text{\tiny|}} \text{dec}(t_n)$ using the token definitions. Any string $\boldsymbol{\sigma} = c_1 \mathbin{\text{\tiny|}} \cdots \mathbin{\text{\tiny|}} c_m \in \Sigma^*$, $m \in \mathbb{N}$ given to the encoder is tokenized greedily from left to right, each time selecting the token in the vocabulary that represents the most characters starting from the beginning of the string. Specifically, the first token in $\text{enc}(\boldsymbol{\sigma})$ is the token $t \in V$ such that $\text{dec}(t) = c_1 \mathbin{\text{\tiny|}} \cdots \mathbin{\text{\tiny|}} c_i$, with $i \leq m$, and $\nexists t' \in V$ such that $\text{dec}(t') = c_1 \mathbin{\text{\tiny|}} \cdots \mathbin{\text{\tiny|}} c_j$ with $i < j \leq m$. In the above selection, if $c_1$ is inside a word, then $t$ must contain the special prefix. This process continues in the same manner with the remaining string $c_{i+1} \mathbin{\text{\tiny|}} \cdots \mathbin{\text{\tiny|}} c_m$.

**Theorem B.2.** *Let $\mathcal{T} = (\Sigma, V, \text{enc}, \text{dec})$ be a Wordpiece-based tokenizer and $\mathbf{s} \in V^*$ a non-canonical token sequence according to $\mathcal{T}$. Then, for any token $t \in V$, $\mathbf{s} \mathbin{\text{\tiny|}} t$ is also non-canonical.*

*Proof.* If $\mathbf{s} = t_1 \mathbin{\text{\tiny|}} \cdots \mathbin{\text{\tiny|}} t_n$ is non-canonical according to Wordpiece, then let $\mathbf{s}' = t'_1 \mathbin{\text{\tiny|}} \cdots \mathbin{\text{\tiny|}} t'_{n'}$ denote the canonical tokenization of the same character string, $\text{dec}(\mathbf{s}) = \text{dec}(\mathbf{s}')$, $n, n' \in \mathbb{N}$. Because $\mathbf{s}' \neq \mathbf{s}$, there must exist at least one $i \leq \min(n, n')$ such that $t_i \neq t'_i$. It is impossible that $t_i = t'_i$ for all $i \in \min(n, n')$ but $n \neq n'$, because then $\text{dec}(\mathbf{s}) \neq \text{dec}(\mathbf{s}')$, as one would be a prefix of the other. Let $t_i, t'_i$, with $i \in \min(n, n')$ be the first different token between $\mathbf{s}$ and $\mathbf{s}'$, *i.e.*, $\forall j < i : t_j = t'_j$ but $t_i \neq t'_i$. Since $\mathbf{s}'$ is canonical, it must be that $|t'_i| > |t_i|$, where $|t| = |\text{dec}(t)|$ represents the size of token $t$ based on how many characters in $\Sigma$ it encodes. Because $\mathbf{s}$ is a prefix of $\mathbf{s} \mathbin{\text{\tiny|}} t$, the first $i$ tokens are the same, but $\mathbf{s} \mathbin{\text{\tiny|}} t$ cannot be canonical because at (token) index $i$ there exists $t'_i \in V$ that encodes more characters than $t_i$, $|t'_i| > |t_i|$. $\qquad\square$

# C. Proof of Theorem 3.2

Here, we provide the proof of Theorem 3.2, which we restate below.

**Theorem 3.2.** Let $d$ be absolutely continuous with respect to $p$. Moreover, assume that there exist $\mathbf{s} \in V^*$ and $t_1, t_2 \in V$ such that $\mathbf{s} \mid t_1$ is non-canonical with $d(\mathbf{s} \mid t_1) > 0$ and $\mathbf{s} \mid t_2$ is canonical with $p(\mathbf{s} \mid t_2) > 0$ and $d(\mathbf{s} \mid t_2) > 0$. Then, it holds that

$$\mathrm{KL}(p, \tilde{d}) < \mathrm{KL}(p, d). \tag{4}$$

*Proof.* Assume there exists $\hat{\mathbf{s}} \in V^*$, $t_1, t_2 \in V$ such that $\hat{\mathbf{s}} \mid t_1$ is non-canonical and $d(\hat{\mathbf{s}} \mid t_1) > 0$ and $\hat{\mathbf{s}} \mid t_2$ is canonical and $p(\hat{\mathbf{s}} \mid t_2) > 0$ and $d(\hat{\mathbf{s}} \mid t_2) > 0$. Given any token sequence $\mathbf{s} \in V^*$, let $p_{\mathbf{s}} = P[T \mid \mathbf{S} = \mathbf{s}]$ be the true next token distribution and $d_{\mathbf{s}}, \tilde{d}_{\mathbf{s}}$ be the next token distribution and canonicalized next token distribution given by the LLM. Then, $d_{\hat{\mathbf{s}}}(t_1) > 0$. Then, we have that $Z = \sum_{t \in V: \, \hat{\mathbf{s}} \mid t \text{ is canonical}} d_{\hat{\mathbf{s}}}(t) < 1$. By definition of $\tilde{d}_{\hat{\mathbf{s}}}$, this implies that for all $t \in V$ such that $\tilde{d}_{\hat{\mathbf{s}}}(t) > 0$, we have that

$$\frac{\tilde{d}_{\hat{\mathbf{s}}}(t)}{d_{\hat{\mathbf{s}}}(t)} > 1 \tag{5}$$

Note that, because $\hat{\mathbf{s}}$ is canonical (by Theorem 3.1) and $d(\hat{\mathbf{s}}) > 0$, it implies that $\hat{\mathbf{s}}$ also has positive probability under $\tilde{d}$, *i.e.*, $\tilde{d}(\hat{\mathbf{s}}) > 0$. In particular, by definition of $\tilde{d}$ we know that $\tilde{d}(\hat{\mathbf{s}})/d(\hat{\mathbf{s}}) \geq 1$ and thus using Eq. 5 it follows that for any $t$ such that $\tilde{d}_{\hat{\mathbf{s}}}(t) > 0$, $\tilde{d}(\hat{\mathbf{s}} \mid t) > 0$ and

$$\frac{\tilde{d}(\hat{\mathbf{s}} \mid t)}{d(\hat{\mathbf{s}} \mid t)} = \frac{\tilde{d}(\hat{\mathbf{s}})}{d(\hat{\mathbf{s}})} \cdot \frac{\tilde{d}_{\hat{\mathbf{s}}}(t)}{d_{\hat{\mathbf{s}}}(t)} > 1 \tag{6}$$

We show that the difference in KL divergence of $p$ from $d$ and $p$ from $\tilde{d}$ is greater than zero. First, consider that we can rewrite the difference as follows

$$\mathrm{KL}(p, d) - \mathrm{KL}(p, \tilde{d}) = \sum_{\mathbf{s} \in V^*} p(\mathbf{s}) \log \left( \frac{p(\mathbf{s})}{d(\mathbf{s})} \right) - \sum_{\mathbf{s} \in V^*} p(\mathbf{s}) \log \left( \frac{p(\mathbf{s})}{\tilde{d}(\mathbf{s})} \right)$$

$$= \sum_{\mathbf{s} \in V^*} p(\mathbf{s}) \left[ \log \left( \frac{p(\mathbf{s})}{d(\mathbf{s})} \right) - \log \left( \frac{p(\mathbf{s})}{\tilde{d}(\mathbf{s})} \right) \right]$$

$$= \sum_{\mathbf{s} \in V^*} p(\mathbf{s}) \log \left( \frac{\tilde{d}(\mathbf{s})}{d(\mathbf{s})} \right) \tag{7}$$

$$= \sum_{\mathbf{s} \in V^*: \, \tilde{d}(\mathbf{s}) > 0} p(\mathbf{s}) \log \left( \frac{\tilde{d}(\mathbf{s})}{d(\mathbf{s})} \right) \tag{8}$$

where the first equations follow from simple manipulations and Eq. 8 follows from the following argument. Whenever $\tilde{d}(\mathbf{s}) = 0$, it implies that either $d(\mathbf{s}) = 0$ or that $\mathbf{s}$ is non-canonical. In both cases, it implies that $p(\mathbf{s}) = 0$ (either by absolute continuity or non-canonicity). Whenever $p(\mathbf{s})$ and $\tilde{d}(\mathbf{s})$ is zero, the contribution of the corresponding term in Eq. 7 is interpreted as zero because $\lim_{x \to 0^+} x \log x = 0$.

We can break up Eq. 8 into two types of summand. For any $\mathbf{s} \neq \hat{\mathbf{s}} \mid t, t \in V$ and $\tilde{d}(\mathbf{s}) > 0$, it readily follows from the definition of $\tilde{d}_{\mathbf{s}}$ that

$$p(\mathbf{s}) \ln \left( \frac{\tilde{d}(\mathbf{s})}{d(\mathbf{s})} \right) \geq p(\mathbf{s}) \ln(1) = 0$$

For any $\mathbf{s} = \hat{\mathbf{s}} \mid t, t \in V$ and $\tilde{d}(\mathbf{s}) > 0$ and $p(\mathbf{s}) > 0$, it follows from Eq. 6 that

$$p(\mathbf{s}) \ln \left( \frac{\tilde{d}(\mathbf{s})}{d(\mathbf{s})} \right) > p(\mathbf{s}) \ln(1) = 0$$

Thus, we can conclude that as there exist $t_2 \in V$ such that $p(\hat{\mathbf{s}} \mid t_2) > 0$ and $\tilde{d}(\hat{\mathbf{s}} \mid t_2) > 0$,

$$\mathrm{KL}(p, d) - \mathrm{KL}(p, \tilde{d}) > 0.$$

$\square$

## Proof of Equation 3

$$\mathrm{KL}(p_{\mathrm{dec}}, \tilde{d}_{\mathrm{dec}})$$

$$= \sum_{\boldsymbol{\sigma} \in \Sigma^*} p_{\mathrm{dec}}(\mathrm{dec}(\mathbf{s}) = \boldsymbol{\sigma}) \ln \left( \frac{p_{\mathrm{dec}}(\mathrm{dec}(\mathbf{s}) = \boldsymbol{\sigma})}{\tilde{d}_{\mathrm{dec}}(\mathrm{dec}(\mathbf{s}) = \boldsymbol{\sigma})} \right)$$

$$= \sum_{\mathrm{enc}(\boldsymbol{\sigma}), \boldsymbol{\sigma} \in \Sigma^*} p(\mathbf{s} = \mathrm{enc}(\boldsymbol{\sigma})) \ln \left( \frac{p(\mathbf{s} = \mathrm{enc}(\boldsymbol{\sigma}))}{\tilde{d}(\mathbf{s} = \mathrm{enc}(\boldsymbol{\sigma}))} \right)$$

$$= \sum_{\mathbf{s} \in V^* : \mathbf{s} \text{ is canonical}} p(\mathbf{s} = \mathbf{s}) \ln \left( \frac{p(\mathbf{s} = \mathbf{s})}{\tilde{d}(\mathbf{s} = \mathbf{s})} \right)$$

$$= \mathrm{KL}(p, \tilde{d})$$

