# OpenReview forum: "Canonical Autoregressive Generation"
_ICML.cc/2025/Workshop/TokShop — TokShop_

### Official Review · Reviewer_AfaJ · 2025-06-08
**A theoretical sound approach for Canonical tokenization**

**Rating:** 7
**Confidence:** 2

**Review:**

This paper formulates and articulates the solution to the previously under-explored challenge of non-canonical token sequence generation in LLMs. It has applicable practical relevance in the areas of model safety, financial transparency, as well as metric measurement.

Quality & Clarity: The theoretical strength is in the newly developed theorem (Theorem 3.1), which formulates the need for canonical token sequences in the sequence as a whole to be such. The intuitive observation, and extension to commonly used tokenizers like BPE, Unigram, and Wordpiece, is persuasive. The proposed canonical sampling technique is intuitive but effective, and the efficiency with which it is carried out using the Gumbel-Max trick is a natural strength, with practical viability. The paper is well structured, with adequate explanation of important concepts like tokenizers as well as the autoregressive generation process.


Originality & Implication: The paper's main contribution to the fact that canonical sampling has the demonstrated effect of moving the resulting token sequence distribution closer to the training distribution (in terms of KL-divergence) is perceived to be a valuable theoretical contribution. This has clear implications in terms of model fidelity and adherence to training data. While the earlier work effectively talked about the influence of non-canonical tokenizations, the present work provides a simple solution to avoid them.


Pros

It addresses the significant and novel problem with genuine negative side effects to LLMs.
Sound theoretical foundation provided by the central theorem about the partial canonical sequences and its supposed generalization to different tokenizers.

Provably enhancing the fidelity of the distributions over token sequences generated from the training distribution.

This sampling plan is efficient as well as easy to implement using the technique of the Gumbel-Max trick.

Cons


A significant restriction the authors acknowledge is that provably useful improvements in token sequence distribution do not imply corresponding improvements in output string distribution. As end-users derive utility from the output strings, empirical verification of improvement at the string level is applicable. In response to the surprising extension to Wordpiece, Unigram, and the BPE tokenizer in the main theorem, the paper suggests enhanced theoretical insight into the properties required in such an extension to further fortify the work.

Lack of Empirical Testing: The research is theoretical in nature. Most notably, it is devoid of any empirical test to show the real effect of canonical sampling in natural generation applications or in preventing the negative effect it is attempting to rectify. It has a most critical lack in demonstrating the real-world value of the technique.


Overall, the paper is an important theoretical and methodological advancement towards more controlled and anticipatable LLM generation. Empirical testing in practice with real-world LLMs and their decoded output strings would be the necessary follow-up step to fully establish its real-world impact.

---

### Official Review · Reviewer_HBzL · 2025-06-08
**Canonical Autoregressive Generation**

**Rating:** 6
**Confidence:** 4

**Review:**

This paper addresses the problem of large language models (LLMs) generating "non-canonical" token sequences, which are tokenizations that deviate from their tokenizer's unique, deterministic output. Its first novel contribution is a theorem proving that once a generated sequence becomes non-canonical, any extension of it will also remain non-canonical. Building on this insight, the authors introduce "canonical sampling," a simple and efficient method using the Gumbel-Max trick to constrain LLM generation to only canonical sequences at each step. The paper's main contribution is a formal proof that this new sampling strategy produces a token distribution provably closer (in KL-divergence) to the true training data distribution. This work is significant for improving LLM reliability and mitigating issues in safety and evaluation caused by tokenization ambiguity. The paper exhibits strong methodological rigor and technical depth with formal proofs extending to BPE, Unigram, and Wordpiece tokenizers. However, the work is purely theoretical and provides no experimental validation to demonstrate the practical performance benefits or computational overhead of the proposed method.

Strengths:
1. The paper clearly articulates a subtle but significant problem of non-canonical token generation and provides a compelling motivation by linking it to practical issues in safety, evaluation, and economic incentives.
2. It introduces a novel theoretical result (Theorem 3.1) which formally establishes that non-canonicality is an irreversible property during generation, providing a solid foundation for the proposed solution.
3. The proposed "canonical sampling" is both simple and practical, cleverly employing the Gumbel-Max trick to efficiently enforce canonicity without the prohibitive computational cost of exhaustive checks.
4. The paper provides a strong theoretical justification for its method by proving that the resulting distribution over token sequences is closer (in terms of KL-divergence) to the true data distribution.
5. The core theoretical claims are generalized to multiple major tokenization schemes (BPE, Unigram, Wordpiece), and the authors are transparent about the limitations of their findings, particularly the token-level vs. string-level distinction.

Weakness:
1. The paper's most significant weakness is its complete lack of experiments, leaving the practical utility, computational overhead, and impact on generation quality of canonical sampling entirely unevaluated.
2. The paper motivates its work by describing the harms of non-canonical generation but provides no data on its prevalence, making it difficult to assess the practical severity of the problem in modern LLMs.
3. The core theoretical guarantee of being closer to the true distribution is in token space, which, as the authors admit, does not necessarily translate to an improvement in the more crucial string-level distribution.
4. By constraining the model's output space, the method might have unintended negative consequences on generation quality, such as reducing creativity or stylistic diversity, which are not investigated.
5. The efficiency of the proposed algorithm relies on a black-box is canonical check, but the paper does not analyze the computational complexity of this check itself, which could be a non-trivial bottleneck.

---

### Decision · Program_Chairs · 2025-06-10

Accept